# Autophagy Alters Bladder Angiogenesis and Improves Bladder Hyperactivity in the Pathogenesis of Ketamine-Induced Cystitis in a Rat Model

**DOI:** 10.3390/biology10060488

**Published:** 2021-05-30

**Authors:** Jian-He Lu, Yi-Hsuan Wu, Tai-Jui Juan, Hung-Yu Lin, Rong-Jyh Lin, Kuang-Shun Chueh, Yi-Chen Lee, Chao-Yuan Chang, Yung-Shun Juan

**Affiliations:** 1Emerging Compounds Research Center, Department of Environmental Science and Engineering, College of Engineering, National Pingtung University of Science and Technology, Pingtung 91201, Taiwan; toddherpuma@yahoo.com.tw; 2Department of Urology, College of Medicine, Kaohsiung Medical University, Kaohsiung 80708, Taiwan; maivy0314@gmail.com; 3Department of Urology, Kaohsiung Medical University Hospital, Kaohsiung 80756, Taiwan; spacejason69@yahoo.com.tw; 4Graduate Institute of Clinical Medicine, College of Medicine, Kaohsiung Medical University, Kaohsiung 80708, Taiwan; 5Department of Medicine, National Defense Medical College, Taipei 11490, Taiwan; u9181002@gmail.com; 6School of Medicine, College of Medicine, I-Shou University, Kaohsiung 84001, Taiwan; ed100464@edah.org.tw; 7Division of Urology, Department of Surgery, E-Da Cancer Hospital, Kaohsiung 82445, Taiwan; 8Division of Urology, Department of Surgery, E-Da Hospital, Kaohsiung 82445, Taiwan; 9Department of Parasitology, College of Medicine, Kaohsiung Medical University, Kaohsiung 80708, Taiwan; rjlin@kmu.edu.tw; 10Graduate Institute of Medicine, College of Medicine, Kaohsiung Medical University, Kaohsiung 80708, Taiwan; yichen83@kmu.edu.tw (Y.-C.L.); chaoyuah@kmu.edu.tw (C.-Y.C.); 11Department of Urology, Kaohsiung Municipal Ta-Tung Hospital, Kaohsiung 80661, Taiwan; 12Department of Anatomy, School of Medicine, College of Medicine, Kaohsiung Medical University, Kaohsiung 80708, Taiwan

**Keywords:** bladder, ketamine, ulcerative cystitis, autophagy, angiogenesis

## Abstract

**Simple Summary:**

Long-term ketamine abuse may increase urinary frequency, nocturia, urgency, bladder pain, dysuria, and sometimes hematuria. Evaluation of the pathophysiological mechanism of bladder voiding dysfunction in ketamine-induced cystitis (KIC) patients is a critical step for therapy. This study uses autophagy inducer (rapamycin, mTOR inhibitor) and inhibitor (wortmannin, PI3K-III inhibitor) to identify the role of autophagy in bladder angiogenesis alteration and bladder hyperactivity improvement.

**Abstract:**

The present study attempts to elucidate whether autophagy alters bladder angiogenesis, decreases inflammatory response, and ameliorates bladder hyperactivity—thereby influencing bladder function in ketamine-induced cystitis (KIC). In our methodology, female Sprague-Dawley (S-D) rats were randomly divided into the control group, the ketamine group, the ketamine+rapamycin group, and the ketamine+wortmannin group. The bladder function, contractile activity of detrusor smooth muscle, distribution of autophagosome and autolysosome, total white blood cells (WBCs) and leukocyte differential counts, the expressions of autophagy-associated protein, angiogenesis markers, and signaling pathway molecules involved in KIC were tested, respectively. The data revealed that treatment with ketamine significantly results in bladder overactivity, enhanced interstitial fibrosis, impaired endothelium, induced eosinophil-mediated inflammation, swelling, and degraded mitochondria and organelles, inhibited angiogenesis, and elevated the phosphorylation of Akt. However, treatment with rapamycin caused an inhibitory effect on vascular formation, removed ketamine metabolites, decreased the eosinophil-mediated inflammation, and ameliorated bladder hyperactivity, leading to improve bladder function in KIC. Moreover, wortmannin treatment reduced basophil-mediated inflammatory response, improved bladder angiogenesis by increasing capillary density and VEGF expression, to reverse antiangiogenic effect to repair KIC. In conclusion, these findings suggested that autophagy could modulate inflammatory responses and angiogenesis, which improved bladder function in KIC.

## 1. Introduction

Clinical studies have revealed that ketamine users may cause severe lower urinary tract symptoms (LUTSs), resulting in increased urinary frequency, nocturia, urgency, bladder pain, dysuria, and sometimes hematuria. These symptoms are very similar to those of interstitial cystitis/bladder pain syndrome (IC/BPS) [1,2,3]. Pathological features by endoscopy are observed in ketamine-addiction patients with bladder erythematous mucosa, mucosa ulceration and laceration, wall thickening, hydronephrosis, and ureter mucosa swelling [4,5]. Clinical ketamine-induced cystitis (KIC) patients show increases in bladder mast cell and eosinophil cell infiltration with increased serum immunoglobulin-E (IgE) levels associated with hypersensitivity and/or allergic reactions [6,7,8]. Besides, pathological changes in the KIC animal model were denuded urothelium, neurogenic inflammation, abnormal apoptosis, bladder wall thickening, and infiltration of mast cells, eosinophils, lymphocytes, as well as plasma cells [8]. Previous evidence suggested the toxic effect of ketamine metabolites result in bladder barrier dysfunction, neurogenic inflammation, immunoglobulin-E-mediated inflammation, and nitric oxide synthase-mediated inflammation, which contribute to the pathogenesis of KIC [8]. Several medical studies showed that the intravesical instillation of hyaluronic acid and botulinum toxin type A were effective for symptom relief in selected patients. However, the pathophysiological mechanism of bladder voiding dysfunction in KIC patients is still elusive.

Autophagy plays an important role in maintaining cellular homeostasis by degrading and recycling damaged cytoplasmic components, including impaired and toxic aggregated proteins, and damaged and dysfunctional organelles. Previous findings have showed that ketamine induces voiding dysfunction by inhibiting calcium influx and smooth muscle contractility [9]. Calcium is a ubiquitous intracellular messenger which affects almost all aspect of cellular life [10], including autophagy [11]. Autophagy is initiated by inhibition of the mammalian target of rapamycin (mTOR) and then phosphorylation of PI3K-III (class III phosphatidylinositol 3-kinase) complex. During the autophagy process, several autophagy-related (ATG) genes regulate autophagosomes formation and maturation. Microtubule-associated protein light chain 3 (LC3) aggregates on the autophagosomal membrane as markers of autophagy [12,13]. Increasing evidence suggests that autophagy is triggered by different stimuli, such as TNF-α [14,15], reactive oxygen species (ROS) [16], nutrient starvation, ischemia [17], inflammatory and metabolic stress [14,18] or mitochondrial dysfunction [19,20]. Whether autophagy plays a protective or a harmful role, it is not clearly established for most diseases. A previous study indicated that the mTOR inhibitor rapamycin reduced bladder overcapacity, smooth muscle hypertrophy, and improved pathologic residual urine volumes, when given during the genesis of obstruction [21]. In vivo mTOR inhibition may attenuate obstruction, induce detrusor hypertrophy and help preserve bladder function. In the cyclophosphamide—treated rats, bladder micturition function was significantly improved after treated with rapamycin. Moreover, bladder inflammation and oxidative stress were lessened [22].

In chronic ketamine abusers, peripheral blood serum vascular endothelial growth factor (VEGF) protein is reduced [23]. The mechanism of angiogenesis by the VEGF signaling pathway is through the VEGF receptor and involves stimulating phosphorylation of Erk1/2, P38, and Akt [24]. Angiogenesis might play a vital role in maintaining blood nutrition and oxygen to supply the regeneration of a dysfunction bladder. Autophagy inducer and inhibitor could be applied as pharmacological agents to regulate angiogenesis. A previous finding inhibition of autophagy by 3-MA ameliorated ischemic brain damage [25] suggested that autophagy reduction is associated with neuroprotection in a cerebral ischemic model. Inhibition of the mTOR pathway by rapamycin prevents cytochrome *c* release and reduces ischemic brain damage in rats subjected to transient forebrain ischemia [26]. Mesenchymal stem cells (MSCs) treated with rapamycin (autophagy inducer) secrete a high level of VEGF via the modulation of Erk phosphorylation and accelerate the regeneration of wound healing. Starvation-induced autophagy triggers cell migration and angiogenesis by activating VEGF and Akt protein on endothelial cells. Zheng et al. reported that the activation of the Akt/eNOS axis inhibited autophagy and improved angiogenesis in cerebral ischemia-reperfusion injury [17].

Rat MSCs with low intensity shock wave therapy were found to promote autophagy through the PI3K/AKT/mTOR pathway. A combination of low intensity shock wave therapy with MSC therapy can express more VEGF than a single treatment, which involves participating in autophagy by triggering the PI3K/Akt/mTOR signaling pathway. This combined therapy could provide a new research direction in erectile dysfunction treatment [27]. To find the relationship between autophagy and angiogenesis is critical in bladder regeneration for KIC. Moreover, whether autophagy is involved in leukocyte-mediated inflammation and angiogenesis, leading to contributing to the pathogenesis of KIC, remains unknown.

In the present study, we hypothesized that rapamycin would have a protective effect by altering angiogenesis and reducing the inflammatory response in the treatment of KIC. The present work was focused on elucidating the antiangiogenic potential of autophagy involved in leukocyte-mediated inflammation and angiogenesis in a rat model of KIC. Understanding molecular mechanisms of autophagy in bladder angiogenesis and leukocyte—mediated inflammation can offer potential therapeutic approaches for overactive bladder (OAB) of ketamine abusers.

## 2. Materials and Methods

### 2.1. Animals and Ketamine Administration

Experiments were performed on 40 female Sprague-Dawley (S-D) rats (Animal Center of BioLASCO, Taipei, Taiwan) weighed between 200 and 250 gm. All experiments using S-D rats were performed in accordance with the guidelines of the Kaohsiung Medical University Institutional Animal Care and Use Committee (IACUC) (IACUC approval number—IACUC-102198, date: 5 April 2014). All animals were maintained at 22 ± 2 °C with 12 h light/dark cycles. S-D rats were distributed into four experimental groups: (1) the control group, which received 0.9% Saline, (2) the ketamine group, which received ketamine (30 mg/kg/day, Pfizer), [28] the ketamine+rapamycin group, which was treated with ketamine in combination with rapamycin (LC Laboratories, 2 mg/kg; autophagy inducer), and (4) the ketamine+wortmannin group, which was treated with ketamine combined with wortmannin (0.5 mg/kg; autophagy inhibitor). All animals received intraperitoneal (IP) injection for 3 months [29,30,31]. Rats were weighed once at the beginning of every week for adjusting the amount of ketamine, rapamycin, or wortmannin to be administered. This study was approved by the Animal Care and Treatment Committee of Kaohsiung Medical University. All experiments were conducted according to the guidelines for laboratory animal care.

After induction of anesthesia with 4% isoflurane, rats were subjected to cardiac puncture and perfused with 0.9% saline solution. Blood samples were collected to analyze the total WBC, individual leukocyte counts, and cell morphology. Blood samples were stayed to clot for 2 h on the ice and centrifuged at 1500*g* × at 4 °C for 15 min to separate out the serum samples for further detection. The bladders were removed, and carefully horizontally dissected into two portions: the upper part of bladder tissue was fixed in 4% paraformaldehyde solution for histological analysis; another part was placed in liquid nitrogen for protein analysis.

### 2.2. Isovolumetric Cystometrograms (CMGs)

Rats were anesthetized with Zoletil-50 (1 mg/kg, IP). Before beginning CMG, the bladder was emptied out, and PE50 urethral catheter was placed. Subsequently, the bladder was injected with 0.9% saline at a 0.08 mL/min rate, and the bladder pressure was measured. CMG was recorded until the bladder pressure was stable, and at least 5 filling/voiding cycles were measured. Pressure and force signals were amplified (ML866 PowerLab, ADInstrument, Colorado Springs, CO, USA), recorded on a chart recorder, and digitized for computer data collection (Labchart 7, ADInstruments: Windows 7 system) [29,31]. CMG parameters recorded for each animal included filling pressure, peak micturition pressure, bladder capacity, and the frequency of non-voiding contractions (without urine leakage during bladder infusion).

### 2.3. Tracing Analysis of Voiding Behavior by Metabolic Cage

After treatment, rats were placed in individual metabolic cages (R-2100; Lab Products, Rockville, Maryland). The 24 h micturition frequency, and voided volume were determined using a cup especially fitted to a transducer (MLT 0380, ADI Instruments, Colorado Springs, CO, USA). The volumes of water intake and urine output were also collected and measured [29,31].

### 2.4. Bladder Contractility Studies

The bladders were removed to record weight and cut open in a sagittal direction. The contractile response was determined in three ways (electrical-field stimulation (EFS), carbachol, and KCl), which were associated with synaptic transmission, receptor responses, or smooth muscle dysfunction, respectively. Three longitudinal strips measuring 0.5 × 1.5 cm^2^ were obtained from the bladder dome to the trigone area, including the urothelial, suburothelial, and muscular layers. Contractile responses were measured as previously described [31,32,33,34]. An initial resting tension of 2 g was applied for 30 min, and contractile responses were recorded isometrically using a force-displacement transducer. At this length, the responses to field stimulation at 2, 8, as well as 32 Hz, and to carbachol (20 μM) and KCl (120 mM) were assessed. For each form of stimulation, only the maximum tension was recorded. All data were recorded on a Grass model D polygraph (Grass Instruments, Warwick, RI, USA), and the data were digitized and analyzed using the Grass POLYVIEW A-D and conversion system (Grass Instrument Co., Warwick, RI, USA).

### 2.5. Ketamine Metabolites Assay in Urine and Serum

Blood was separated by centrifugation at 4 °C, and serum was collected to analyze ketamine and its major metabolite norketamine. Additionally, the urine was also collected by the metabolic cage. The concentrations of ketamine and norketamine in urine and serum were determined by using a modified version of the high-performance liquid chromatography (HPLC). After extraction and purification by liquid–liquid extraction using ethyl ether, urine samples underwent chromatography on a reversed-phase column, and ketamine and norketamine were detected at 200 nm by UV spectrophotometry. This study was carried out according to ISO 9001:2000 requirements.

### 2.6. Histological Study by Masson’s Trichrome Stain

Masson’s trichrome stain was performed to investigate the pathological changes in the bladder. S-D rats were perfused with 0.9% saline solution and the bladders were removed to record weight. The removed bladders were fixed overnight, embedded in paraffin blocks, and 5 μm thick sections were obtained. Deparaffinized sections were stained with Masson’s trichrome stain (Masson’s Trichrome Stain Kit HT15, Sigma, St. Louis, MO, USA). A standard Masson’s trichrome staining procedure was followed [29,30,31,35] to stain the connective tissue in blue and detrusor smooth muscle (DSM) in red. Histology slides stained with Masson’s Trichrome were inspected and evaluated by two independent pathologists.

### 2.7. Measurement of Leukocyte Count and Blood Smear

The objectives of the hematology analyzer and blood smear were to evaluate total white blood cells (WBCs), individual leukocyte counts, and cell morphology. Count analysis of leukocytes was carried out by the hematology analyzer based on flow cytometry (Model SF-3000, Sysmex Co., Kobe, Japan). The analyzer could fractionate neutrophils, lymphocytes, monocytes, eosinophils, and basophils. The percentage between individual leukocytes and total WBCs was calculated [36]. Moreover, whole blood samples were smeared using a thin layer of blood on slides and stained with a modified Wright-Giemsa stain (Sigma, WG32) to allow the various blood cells to be examined microscopically. When blood slides were stained using Wright-Giemsa Stain, the white blood cell nucleus and cytoplasm took on the characteristic blue or pink coloration.

### 2.8. Transmission Electron Microscopy (TEM)

TEM was used to investigate the ultrastructure of bladder tissues. The bladder tissues were harvested and immersed in 0.1 M phosphate-buffered saline containing 2% paraformaldehyde and 2.5% glutaraldehyde overnight. Then, the tissues were fixed with 2% osmium tetroxide at 4 °C for 90 min, embedded in the epon, and sectioned in 70 nm using an automatic ultramicrotome (Reichert Ultracut E, Vienna, Austria). Then, the morphology was examined under a transmission electron microscope (JEM2000 EXII; Jeol Ltd., Tokyo, Japan).

### 2.9. Protein Isolation and Western Blot Analysis

Frozen tissue samples of the bladder were homogenized on ice in lysis buffer containing halt protease inhibitor cocktail (Pierce, Rockford, IL, USA). Protein (16 μg) from the bladders was loaded onto SDS-PAGE gels and transferred to polyvinylidene fluoride (PVDF) membranes (Millipore). The membranes were incubated with primary antibodies, including mTOR (Cell Signaling, rabbit monoclonal IgG, 1:1000, MW: 289 kDa, catalog no. 2983), Phospho-mTOR (p-mTOR; Cell Signaling, rabbit monoclonal IgG, 1:1000, MW: 289 kDa, catalog no. 5536), ATG12 (Proteintech, rabbit polyclonal IgG, 1:1000, MW: 48–55 kDa, catalog no. 11264-1-AP), ATG7 (Cell Signaling, rabbit monoclonal IgG, 1:1000, MW: 78 kDa, catalog no. 8558), Beclin 1 (ATG6; Novus, rabbit polyclonal IgG, 1:1000, MW: 52–55 kDa, catalog no. NBP1-76648), LC3A (ATG8; Cell Signaling, rabbit monoclonal IgG, 1:1000, MW: 14 and 16 kDa, catalog no. 4599), VPS 34 (Proteintech, rabbit polyclonal IgG, 1:1000, MW: 100 kDa, catalog no. 12452-1-AP), alpha-smooth muscle actin (α-SMA; Abcam, rabbit monoclonal IgG, 1:5000, MW: 40 kDa, catalog no. ab5694), CD31 (endothelial marker; Abcam, mouse monoclonal IgG, 1:3000, MW: 83 kDa, catalog no. 9498), vascular endothelial growth factor (VEGF; Millipore, mouse monoclonal IgG, 1:2000, MW: ∼40 kDa, catalog no. 05443), VEGF-R1 (VEGF receptor; Abcam, rabbit monoclonal IgG, 1:1000, MW: 150 kDa, catalog no. ab32152), VEGF-R2 (VEGF receptor; Cell Signaling, rabbit monoclonal IgG, 1:1000, MW: 220 kDa, catalog no. #9698), Laminin (Abcam, rabbit monoclonal IgG, 1:1000, MW: 200–400 kDa, catalog no. ab11575), integrin-α6 (Laminin receptor; Abcam, rabbit monoclonal IgG, 1:5000, MW: ∼127 kDa, catalog no. ab181551), Erk1/2 (p44/42; Cell Signaling, rabbit monoclonal IgG, 1:1000, MW: 42–44 kDa, catalog no. #9102), Phospho-Erk1/2 (p-Erk1/2; Cell Signaling, rabbit monoclonal IgG, 1:2000, MW: 42–44 kDa, catalog no. #4370), p38 (Cell Signaling, rabbit monoclonal IgG, 1:1000, MW: 38–40 kDa, catalog no. #8690), Phospho-P38 (p-P38; Cell Signaling, rabbit monoclonal IgG, 1:1000, MW: 38–40 kDa, catalog no. #4511), Akt (Cell Signaling, mouse monoclonal IgG, 1:2000, MW: 60 kDa, catalog no. #2920), Phospho-Akt (p-Akt; Cell Signaling, rabbit monoclonal IgG, 1:1000, MW: 60 kDa, catalog no. #4060), ß-actin (Cell Signaling, rabbit monoclonal IgG, 1:5000, MW: 43 kDa, catalog no. 4970S), and glyceraldehyde-3-phosphate dehydrogenase (GAPDH; Millipore, mouse monoclonal IgG1, 1:1000, MW: 36 kDa, catalog no. MAB374). Therefore, the autophagy-related proteins [mTOR, p-mTOR, ATG12, ATG7, Beclin 1 (ATG6), LC3 (ATG 8), and VPS 34], angiogenesis-related proteins and receptors (α-SMA, CD31, VEGF, VEGF-R1, VEGF-R2, laminin, and integrin-α6) and cell signal-related proteins (Erk1/2, p-Erk1/2, P38, p-P38, Akt, and p-Akt) were normalized with ß-actin and GAPDH. In each experiment, negative controls were also done without primary antibody. Please find the full Western Blot in Appendix A.

### 2.10. Immunofluorence Staining, Confocal Microscopy, and Automated Computer-Based Image Quantification for the Location of Protein Expression

For in vivo bladder section, immunostaining was performed according to published methods [29,30,31]. The bladder whole mounts were fixed in cold 4% paraformaldehyde in 0.1 M PBS overnight, washed with PBS, blocked with 10% NGS in PBS/0.5% Triton X-100 for 1hr at room temperature (RT), and then incubated with the primary antibody to laminin (Abcam, rabbit polyclonal IgG, 1:100 catalog no. ab11575), α-SMA (Abcam, rabbit monoclonal IgG, 1:100, catalog no. ab5694) and LC3 (Santa Cruz, mouse monoclonal IgG, 1:50, catalog no. sc-376404) at 4 °C over two night. Tissues were washed in PBS/0.5% Triton X-100 15 min on a rocker and incubated with secondary antibodies at RT for 2 h. After washing, DAPI was added, and the whole mounts were transferred to a glass slide and coverslipped with a 100 mm adhesive spacer in Prolong Gold anti-fade reagent (Invitrogen). Double staining of the green-stained laminin and the red-stained LC3 in the bladder whole mount were captured using confocal laser scanning microscopy. In each experiment, negative controls without the primary antibody were performed to elucidate non-specific immunostaining.

### 2.11. Statistical Analysis

Analysis of variance, followed by the Bonferroni test and two-way analysis of variance for individual comparison, was conducted for the above experiments. The mean, standard deviation (SD), and *p* values were calculated on triplicate experiments. A Student’s *t*-test was used to calculate *p*-values for comparison. The significant level was set at a *p*-value < 0.05.

## 3. Results

### 3.1. General Characteristic Evaluation

The physical indicators are shown in Table 1, including water intake, urine output, body weight, bladder weight, and the ratio of bladder and body weight. There was no significant difference in the amount of water intake, urine output, and body weight among different groups. Additionally, the bladder weight of the ketamine group and the ketamine+wortmannin group was significantly increased compared to the control group. However, the bladder weight of the ketamine+rapamycin group significantly reduced as compared with the ketamine group.

### 3.2. Effects of Rapamycin Treatment on Improving Bladder Capacity and Voiding Function

Cystometrogram (CMG) parameters, including peak micturition pressure, micturition frequency, interval, voiding, and nonvoiding contractions, are shown in Figure 1 and Table 2. The CMG data of the control group showed regular and stable micturition patterns. Moreover, the ketamine group showed bladder hyperactivity with increases in voiding contraction (arrows), nonvoiding contraction (asterisks), micturition frequency, and peak micturition pressure as compared with the control group. In contrast, both the ketamine+rapamycin group and the ketamine+wortmannin group significantly reduced micturition pressure, as well as micturition frequency, and increased bladder capacity as compared with the ketamine group (Figure 1 and Table 2).

Additionally, tracing analysis of voiding behavior by metabolic cage revealed that the ketamine group decreased voiding volume and increased micturition frequency as compared with the control group (Figure 1B and Table 2). There was an increase in the micturition interval and the voiding volume in the ketamine+rapamycin group, as compared with those in the ketamine group and the ketamine+wortmannin group. Therefore, the micturition function was reduced in the ketamine group, whereas significantly improved in the ketamine+rapamycin group and the ketamine+wortmannin group. Taken together, the above findings revealed that ketamine treatment deteriorated bladder capacity and voiding function, whereas ketamine combined with rapamycin or wortmannin treatment significantly ameliorated KIC-induced bladder overactivity.

### 3.3. Contractile Responses of Bladder Strips after Treatment

Contractile responses of bladder strips in response to electrical-field stimulation (Figure 2A), carbachol (Figure 2B), and KCl (Figure 2C) stimulation are shown in Figure 2 and Table 2. The ketamine group exhibited stronger contractile responses to EFS at 8 and 32 Hz than the control group or the ketamine+rapamycin group (Figure 2A,D and Table 2). Similar observations were found in response to carbachol (Figure 2B,E and Table 2) and KCl (Figure 2C,E and Table 2). Therefore, ketamine administration strongly induced bladder contractile in response to these stimulations as compared with the control group. In contrast, the ketamine+rapamycin group alleviated bladder contractile as compared with the ketamine group. These results suggested that ketamine treatment caused significant bladder contractile response, whereas ketamine combined with rapamycin treatment improved such changes.

### 3.4. Autophagy Inducer and Inhibitor Exhibited No Effect on Ketamine Metabolism

Ketamine was metabolized to norketamine and excreted in the urine. Significant amounts of ketamine and norketamine were found in urine (Table 1). The serum concentrations of ketamine and norketamine were undetectable in all groups. In contrast, the concentrations of urine ketamine and norketamine in the ketamine group were 1268.3 ± 231.5 ng/mL and 10,150.2 ± 1146.8 ng/mL, respectively. In the ketamine+rapamycin group, the concentrations of urine ketamine and norketamine were 1096.7 ± 175.8 ng/mL and 8790.0 ± 1235.4 ng/mL, respectively. In the ketamine+wortmannin group, the concentrations of urine ketamine and norketamine were 1096.7 ± 175.8 ng/mL and 8790.0 ± 1235.4 ng/mL, respectively. The urinary concentrations of ketamine and norketamine were very high after ketamine treatment. There was no significant difference in the urinary concentrations of ketamine and norketamine among the ketamine group, the ketamine+rapamycin group, and the ketamine+wortmannin group. These results revealed that autophagy inducer (rapamycin) and inhibitor (wortmannin) exhibited no effect on ketamine metabolism.

### 3.5. Rapamycin Treatment Improved Ketamine-Associated Bladder Damage and Interstitial Fibrosis

The histopathological features of ketamine-induced bladder damage are shown in Figure 3. There were 3∼5 layers in the urothelial layer (UL; mucosa), while sparse collagen (yellow arrow) in the suburothelial layer (SL; lamina propia) in the control group (Figure 3A,A’).

On the contrary, the morphology of the ketamine group was characterized by ulcerated and denuded urothelium (black arrow), erythrocyte accumulation (hemorrhages; black arrowheads), mononuclear cell infiltration (yellow arrowheads), and increased interstitial fibrosis (yellow arrow) in SL (Figure 3B,B’). Additionally, morphological evaluation of the ketamine+rapamycin group showed improved ketamine-associated bladder damages by reducing ulcerated urothelium, erythrocyte accumulation, mononuclear cell infiltration (yellow arrowheads), and interstitial fibrosis (yellow arrow) (Figure 3C,C’). In the ketamine+wortmannin group, there were many mononuclear cell accumulation (yellow arrowhead), denuded urothelium (black arrow), and interstitial fibrosis (yellow arrow) in SL (Figure 3D,D’), but such pathological damage was not as profound as in the ketamine group.

Based on the histopathology, our data showed that ketamine treatment caused inflammation of bladder mucosa and stroma by ulcerated and denuded urothelium, mononuclear cell accumulation, and increased interstitial fibrosis. Whereas, ketamine combined with rapamycin treatment attenuated these changes. These observations revealed that treatment in the ketamine group significantly exacerbated bladder damage and interstitial fibrosis. In contrast, rapamycin treatment reduced fibrotic biosynthesis and ameliorated bladder damage in KIC.

### 3.6. The Presence of Autophagosome and Autolysosome in Bladder Tissue after Treatment

To directly visualize autophagy, TEM was used to examine autophagic vacuoles (autophagosomes). The ultrastructure of bladder urothelium in UL showed intact nucleus (N) and mitochondria (yellow arrows) in the control group (Figure 3E,E’). Rare autophagosome-like structure was observed in the control group. However, in the ketamine group, we observed damaged urothelium (white arrowheads), many swelling, as well as degraded mitochondria (yellow arrows) and organelles (Figure 3F,F’).

On the contrary, many intact nucleus (N) and mitochondria (arrows) were presented in the ketamine+rapamycin group. Moreover, many damaged cytoplasmic components and degraded organelles were engulfed by double-membrane autophagosome and autolysosome (yellow arrowheads) as compared with the ketamine group (Figure 3G,G’). Some swelling and degraded mitochondria (yellow arrows) were also observed in the ketamine+wortmannin group (Figure 3H,H’).

Besides, the ultrastructure of smooth muscle cells (SMCs) in the muscular layer (ML) showed intact nucleus (N) and mitochondria (yellow arrows) in the control group (Figure 3I,I’). While, autophagic vacuoles were rarely detected in the control group. However, in the ketamine group, the nuclear condensation and shrinkage, and any swelling, as well as degraded mitochondria (yellow arrows) and some degraded organelles engulfed by autolysosome (yellow arrowheads) were observed (Figure 3J,J’).

Whereas, in the ketamine+rapamycin group, intact mitochondria (arrows) and degraded cytoplasmic components and organelles were engulfed by autolysosome (yellow arrowhead) as compared with the ketamine group and the ketamine+wortmannin group (Figure 3K,K’). Intact nucleus (N) and some swelling mitochondria (yellow arrows) were also observed in the ketamine+wortmannin group (Figure 3L,L’). Nevertheless, the administration of rapamycin improved the ultrastructure of the urothelial and muscular layer in the bladders.

Bladder SL and ML contained a rich vascular network, extracellular matrix (ECM), SMCs, sensory nerve endings, and several types of cells, including interstitial cells (ICs) and fibroblasts. The ICs lying in proximity to SMCs (M) and nerve ending (nerve bundle; black arrows) were detected by TEM (Figure 3M–P). In the ketamine group (Figure 3N), the numbers of degraded IC and damaged nerve ending between smooth muscle bundles were higher than the control group (Figure 3M) and the ketamine+rapamycin group (Figure 3O). Besides, the expression of ICs was noticeably enhanced in the ketamine+rapamycin group, as compared with the ketamine group and the ketamine+wortmannin group. These observations suggested that degraded nerve ending, as well as ICs and damaged mitochondria in SL and ML may disturb the activity of smooth muscle contractility to cause the micturition dysfunction and overactivity bladder after treated ketamine.

### 3.7. Protein Expressions of Autophagy-Related Proteins in Bladder

To determine whether the autophagy pathway was activated after treatment, the levels of autophagy-associated proteins [mTOR, p-mTOR, ATG12, ATG7, LC3 (ATG8), Beclin 1 (ATG6), VPS 34] were measured by western blot analysis in bladder tissue. In comparison with the control group, ketamine and its metabolites evoked mTOR phosphorylation (8.0-fold) and up-regulated significantly Beclin1 (1.6-fold), as well as LC3 (1.45-fold) protein in the bladder in the ketamine group. However, in the ketamine+rapamycin group, the levels of autophagy-related proteins (ATG12 (4.5-fold), ATG 7 (17.2-fold), LC3 (4.7-fold), Beclin 1 (6.3-fold), VPS 34 (6.3-fold)), except mTOR and p-mTOR protein, were noticeably enhanced compared to the ketamine group (Figure 4). These findings revealed that rapamycin inhibited ketamine-activated mTOR expression and elevated the level of autophagy-associated proteins.

### 3.8. Significant Changes in Total White Blood Cells (WBCs) and Leukocyte Differential Counts

Leukogram evaluation provides valuable information about disorder, inflammatory conditions, and stress. As shown in Figure 5A and Table 2, the number of total WBCs was significantly increased in the ketamine group and the ketamine+rapamycin group, as compared to the control group and the ketamine+wortmannin group. The percentage of five type leucocytes to total WBCs was present in Figure 5B–F and Table 2. In the control group, the percentages of neutrophils, lymphocytes, monocytes, eosinophils and basophils to total WBCs were 18.77 ± 2.78%, 71.00 ± 7.66%, 7.33 ± 2.20%, 1.90 ± 0.27% and 0.15 ± 0.04% of circulating leukocytes, respectively. The above observations revealed that the number of total WBCs and the percentages of monocytes and eosinophils to total WBCs in the ketamine group were higher than those in the control group (Figure 5A,D,E). Moreover, in the ketamine+rapamycin group, the number of total WBCs and the percentages of neutrophils and basophils to total WBCs were higher than those in the control group and the ketamine group (Figure 5A,B,F). In comparison with the control group versus the ketamine group, the percentages of neutrophils, monocytes, and eosinophils to total WBCs were noticeably enhanced in the ketamine+wortmannin group (Figure 5B,D,E). However, as compared to the control group versus the ketamine group, the percentage of lymphocytes to total WBCs was significantly decreased in the ketamine+rapamycin group and the ketamine+wortmannin group.

According to the ultrastructure of WBCs examined by blood smear and TEM, the prominent feature of mature neutrophils (black arrowheads) is usually 3~5 lobes nucleus. In the present study, the percentage of neutrophils was significantly higher in the ketamine+rapamycin group and the ketakmine+wortmannin group than that in the control group or the ketamine group (Figure 5B,G–J,G’–J’). Lymphocytes (yellow arrowheads) are the smallest cells in total WBCs and are characterized by a round, condensed nucleus and a relatively small amount of cytoplasm. The administration of rapamycin and wortmannin significantly decreased the number of lymphocytes as compared to the control group and the ketamine group (Figure 5C,G–J,G’–J’). Furthermore, monocytes (black arrows) are the largest of the white cells and are characterized by a large, less intense, eccentrically placed nucleus. They are the precursors of macrophages. The number of monocytes was significantly higher in the ketamine group, the ketamine+rapamycin group, and the ketamine+wortmannin group, as compared to the control group (Figure 5D,G–J,G’–J’). The characteristic feature of eosinophils (yellow arrows) is the large, ovoid, specific granules (S), each containing an elongated crystalloid. As shown in Figure 5E,G–J,G’–J’, the number of eosinophils was significantly higher in the ketamine group and the ketamine+wortmannin group, as compared to the control group. However, the administration of rapamycin significantly decreased the percentage of eosinophils to total WBCs.

There are no significant changes in the number of basophils, except in the ketamine+rapamycin group (Figure 5F,G–J,G’–J’). Besides, infiltrated cells of different groups, including lymphocyte, plasma cell (an eccentric round nucleus with condensed chromatin and a well-developed nucleolus) (Figure 5M), macrophage (a large, round cell with a central round nucleus, numerous small pseudopodia extend from the cell and vacuolated cytoplasm) (Figure 5N,R), neutrophil (Figure 5O), eosinophil (Figure 5P), basophil, and mast cell (Figure 5Q) into SL were also shown. The above observations revealed that leucocyte-mediated inflammation induced by plasma cells involving in the induction of activated mast cells may play a significant role in the pathophysiology of KIC.

### 3.9. Autophagy Alters Bladder Angiogenic Remodeling in KIC

To further explore the angiogenic remodeling for bladder repair after treatment with saline (Figure 6A,E), ketamine (Figure 6B,F), rapamycin (Figure 6C,G) and wortmannin (Figure 6D,H), the angiogenesis marker laminin (Figure 6A–D) and α-SMA (Figure 6E–H) immunoreactivity (arrows) were also assessed. The immunostaining was widely distributed in the blood vessel in the urothelial basal layer, SL (lamina propria), and ML of the bladder tissue in the control group (Figure 6A,E). However, the immunoreactivity in the ketamine group (Figure 6B,F) was stained in the blood vessel of the urothelial basal layer and SL. The expression of laminin and α-SMA was much lower in the ketamine+rapamycin group (Figure 6C,G) than in the ketamine group and the control group. Nevertheless, in the ketamine+wortmannin group (Figure 6D,H), their expression was markedly increased in both the UL and SL as compared with the ketamine group and the ketamine+rapamycin group, but the increase was as profound as in the control group.

To evaluate whether autophagy alters bladder angiogenesis in the pathogenesis of KIC, the level of angiogenesis-associated proteins, including α-SMA, CD31, VEGF, VEGF-R1, VEGF-R2, laminin, and integrin-α6 was quantified in bladder UL and ML by western blots (Figure 6I,J). The results showed that the expression of angiogenesis-associated proteins was much lower in the UL (α-SMA (0.4-fold), CD31 (0.3-fold), VEGF (0.6-fold), VEGF-R1 (0.5-fold), VEGF-R2 (0.4-fold), laminin (0.3-fold), integrin-α6 (0.5-fold)) and ML (α-SMA (0.7-fold), CD31 (0.6-fold), VEGF-R1 (0.6-fold), VEGF-R2 (0.6-fold)) of the ketamine group than the control group. However, such level was significantly suppressed as shown in both the UL (α-SMA (0.5-fold), CD31 (0.5-fold), laminin (0.6-fold)) and ML (α-SMA (0.6-fold), CD31 (0.2-fold), VEGF (0.4-fold), VEGF-R1 (0.6-fold), VEGF-R2 (0.4-fold)) of the ketamine+rapamycin group, as compared with the control group, but the decrease was not as profound as in the ketamine group.

Nevertheless, in the ketamine+wortmannin group, their expression was markedly increased in the UL as compared with the ketamine group (α-SMA (2.3-fold), CD31 (9.7-fold), VEGF (8.8-fold), VEGF-R1 (2.7-fold), VEGF-R2 (3.9-fold), laminin (3.9-fold), integrin-α6 (5.9-fold)) and the ketamine+rapamycin group (α-SMA (1.9-fold), CD31 (6.4-fold), VEGF (4.4-fold), VEGF-R2 (1.7-fold), laminin (1.9-fold), integrin-α6 (2.3-fold)), and the increase was as profound as in the control group. Wortmannin has been shown to improve bladder function and blood flow, by increasing capillary density and VEGF expression. The present study suggested that autophagy may be associated with the antiangiogenic potential for bladder repair in the pathogenesis of KIC.

### 3.10. Autophagy Alters Bladder Angiogenesis in the Pathogenesis of KIC

To further elucidate the relationship between autophagy and angiogenesis, the autophagy level in the bladder with rapamycin or wortmannin in KIC was studied. The double-labeled immunofluorescence of laminin (green, yellow arrows, upper right panels) and LC3 (red, yellow arrowheads, bottom left panels) expressions in bladder tissue was performed in the control group (Figure 7A), the ketamine group (Figure 7B), the ketamine+rapamycin group (Figure 7C) and the ketamine+wortmannin group (Figure 7D). The laminin staining was strongly observed in the vessel of UL and SL in the control group, whereas weak LC3 staining was found in SL. Additionally, colabeling of laminin and LC3 was significantly observed in the SL of the ketamine group and the ketamine+rapamycin group, as compared to the control group and the ketamine+wortmannin group.

Moreover, the ultrastructure of the bladder blood vessel (BV) was identified under TEM (Figure 7E–H). The vessel ultrastructure revealed endothelium (E), pericyte (PC), and interstitial cell (IC) with intact nucleus and mitochondria (yellow arrows), which were presented in the blood vessel (BV) of the control group (Figure 7E,E’). In contrast, in the ketamine group (Figure 7F,F’), the damaged endothelium (E) with degraded mitochondria (yellow arrow) was shown. Moreover, intact nucleus, mitochondria (yellow arrows), and autolysosome (yellow arrowhead) were observed in the endothelium (E) of BV in the ketamine+rapamycin group (Figure 7G,G’) and the ketamine+wortmannin group (Figure 7H,H’).

Our investigation revealed that rapamycin treatment interfered with angiogenic potential by enhancing autophagy activity in the endothelium, suggesting that activated autophagy response by rapamycin treatment is a compensatory defensive reaction to eliminate toxic substances of ketamine metabolites in the blood.

### 3.11. Cellular Signaling Pathways of Regulating Autophagy in Angiogenic Remodeling

Elucidating the cellular signaling pathways of regulating autophagy in angiogenic response will benefit KIC treatment and prevention. The level of signaling-related proteins in the UL and ML of the bladder, including Erk1/2, p-Erk1/2, P38, p-P38, Akt, and p-Akt, was quantified by western blots (Figure 8). In UL, the expression level of Erk1/2, p-Erk1/2, P38, p-P38, and Akt proteins was significantly declined in the ketamine group, as compared with the control group (0.4-fold, 0.7-fold, 0.4-fold, 0.2-fold, 0.5-fold, respectively), except p-Akt. The data revealed that treatment with ketamine significantly increased the Akt phosphorylation, but reduced the phosphorylation of Erk1/2 and p38, as compared to the control group.

Moreover, in the ketamine+wortmannin group, the levels of Erk1/2, p-Erk1/2, p38, p-P38, Akt, and p-Akt proteins in the UL all exhibited abundant expression as compared with the ketamine group (2.9-fold, 5.1-fold, 3.0-fold, 6.7-fold, 2.0-fold, 3.0-fold, respectively), whereas no noticeable change was found in the ML. However, treatment with rapamycin increased the p38 phosphorylation, but suppressed the expression of p-Erk1/2 and p-Akt. These observations implied that treatment with ketamine significantly increased the phosphorylation of Akt in UL and ML as compared to the control group, whereas treatment with rapamycin reduced the phosphorylation of Erk1/2 (0.7-fold in UL, 0.4-fold in ML, respectively) and Akt (0.4-fold in UL, 0.1-fold in ML, respectively).

### 3.12. Proposed Potential Mechanism of Antiangiogenesis, Which Was Triggered by Ketamine Metabolite through PI3K/Akt/mTOR Pathway, Contributed to the Pathogenesis of KIC

The proposed potential mechanism of anti-angiogenesis was induced by ketamine, as shown in Figure 9. Our results revealed that ketamine induced autophagy dysregulation, inhibited angiogenesis, and reduced the phosphorylation of Erk1/2 and p38, but increased Akt phosphorylation, leading to impaired urothelium and endothelium, and eventually causing bladder overactivity in S-D rats. The application of autophagy inducer, rapamycin, had an inhibitory effect on ketamine metabolite—induced inflammation and vascular formation through PI3K/Akt/mTOR pathway. Treatment with autophagy inhibitor wortmannin has been shown to improve bladder angiogenesis and blood flow by increasing capillary density and VEGF expression to repair KIC through PI3K/Akt/mTOR pathway and P38 and Erk1/2 MAPK pathway.

## 4. Discussion 

Ketamine-treated rats experienced bladder hyperactivity with increases in peak micturition pressure, micturition frequency, voiding contraction, and non-voiding contraction, but decrease in bladder volume. Morphological evaluation of ketamine-induced bladder damages was shown by reducing ulcerated urothelium, erythematous mucosa, and interstitial fibrosis. However, these changes were ameliorated in the ketamine+rapamycin group and the ketamine+wortmannin group. Moreover, ultrastructure changes of the bladder were detected by TEM in some swelling and degraded mitochondria, as well as nuclear condensation and shrinkage in the ketamine group. However, some degraded mitochondria were engulfed by double-membrane autophagosomes in the ketamine+rapamycin group. Meanwhile, bladder tissues were elevated significantly during the expression of autophagy-associated proteins, including ATG12, ATG7, Beclin1, LC3-I, LC3-II, and VPS 34, in the ketamine+rapamycin group as compared to the ketamine group. Additionally, the expression of angiogenesis-associated markers, including α-SMA, VEGF, VEGF-R1, VEGF-R2, laminin, and integrin-α6, was increased in a urothelial layer in the ketamine+rapamycin group. However, this expression was lessened in the muscular layer (ML) compared to the ketamine group. In addition, KIC induced by the ketamine metabolites resulted in eosinophil-mediated inflammation. Therefore, treatment with rapamycin alleviated the eosinophil-mediated inflammation, leading to improving the pathogenesis of KIC.

Moreover, the level of Akt phosphorylation in ML was increased significantly in the ketamine group, as compared to the control group, whereas the levels of Erk1/2 and p38 phosphorylation were reduced. However, treatment with rapamycin increased the phosphorylation of p38, but reduced the phosphorylation of Akt. Therefore, the application of autophagy inducer, rapamycin, had an inhibitory effect on ketamine metabolite-induced inflammation and vascular formation, while autophagy inhibitor wortmannin improved the anti-angiogenic effect. The above observations implied that treatment with ketamine significantly resulted in bladder overactivity, enhanced interstitial fibrosis, impaired endothelium, inhibited angiogenesis, and elevated the phosphorylation of Akt. However, these changes were ameliorated in the ketamine+rapamycin group. The precise role of autophagy under various conditions may be opposite, differ from protecting cells survival to promote cells death. However, the mechanism is still unclear. 

In our study, the ketamine+rapamycin group significantly reduced peak micturition pressure as well as micturition frequency, and increased bladder capacity as compared with the ketamine group. However, the peak micturition pressures in the ketamine treatment groups are still higher than the control. This might imply, in addition to bladder cell autophagy changes, that there were some other factors affecting bladder micturition pressure, such as pelvic floor hypertonicity induced bladder outlet obstruction or neuroinflammation. The ultrastructural feature was shown by some swelling and degraded mitochondria, as well as damaged cytoplasmic components engulfed by double-membrane autophagosome. Meanwhile, bladder tissues were accompanied by increases in the expression levels of autophagy-associated proteins (ATG12, ATG7, Beclin1, LC3, and VPS 34), which were elevated significantly in the ketamine+rapamycin group as compared to the ketamine group. Besides, treatment with rapamycin alleviated the eosinophil-mediated inflammation to improve the pathogenesis of KIC. The autophagy response was improved after treatment with rapamycin to eliminate toxic substances of ketamine metabolites in the blood vessel and bladder tissue. Therefore, activated autophagy response by rapamycin treatment was considered as a compensatory defensive reaction.

Bladder SL (lamina propria) and ML contain a rich vascular network, elastic fibers, as well as smooth muscle, and are also composed of the extracellular matrix (ECM), sensory nerve endings, and several types of cells, including ICs and fibroblasts. Moreover, urothelial integrity is maintained through a complex process of migration, proliferation, and differentiation. However, in various bladder inflammatory conditions, urothelial cells can accelerate their proliferation. The physiological function of ICs plays important roles in regulating the generation of electrical slow-wave activity, the mediation of neurotransmission between nerve endings and smooth muscle cells, as well as the activity of detrusor muscle contractility. The role of ECM laminin is involved in cell proliferation, adhesion, and migration [37]. Integrin-α6 is a receptor for laminin [38]. They are components of the basement membrane, which mediate adhesion and activate signaling pathways that cooperate with growth factor receptors to regulate bladder function. Mouse genetic studies demonstrated that Integrin-α6 regulated angiogenesis on endothelial cells and macrophages [39]. 

Our results showed that the expressions of laminin and α-SMA were much lower in the UL and ML of the ketamine group than the control group. In addition, the level of angiogenesis-associated proteins, including α-SMA, VEGF-R1, VEGF-R2, laminin, and integrin-α6, was significantly increased in the UL in the ketamine+rapamycin group, as compared with the ketamine group. Nevertheless, in the ketamine+wortmannin group, their expression was markedly increased in the UL, as compared with the ketamine group and the ketamine+rapamycin group, and the increase was as profound as in the control group. Therefore, treatment with wortmannin has been shown to improve bladder function and blood flow, by increasing capillary density and VEGF expression. Based on the above findings, the dual effect of autophagy on anti-angiogenesis that relies on cell type, cellular demand, and condition was suggested.

The diagnosis of KIC should be established by urine ketamine test instead of the only subjective history of ketamine abuser. KIC involves a complex pathophysiological response with epithelial cell damage, endothelial microvasculature dysfunction, and inflammatory response. Significant changes in the number of total white blood cell (WBC) and leukocyte differential counts should be quantified. Pathological changes in the KIC animal model consist of denuded urothelium, collagen accumulation, infiltration of plasma cells, mast cells, eosinophils, and lymphocytes [8]. Clinical KIC patients were shown with increases in bladder mast cell and eosinophil cell infiltration and enhanced serum immunoglobulin-E (IgE) levels, revealing that KIC is associated with hypersensitivity and/or allergic reactions [6,7,8]. Gamper et al. found increased mast cell activation in the bladders of patients with ulcerative interstitial cystitis. The double immunohistochemical staining also showed co-expression of mast cells and IgE. These observations suggested that IgE mast-cell-mediated inflammation may play a significant role in the pathophysiology of KIC [40]. Besides, ketamine might cause an allergic reaction to urothelium and mast cells, which could also play an important role. 

Our results revealed that the number of total WBCs, as well as the percentage of monocytes and eosinophils, was significantly increased in the ketamine group, as compared to the control group. Rapamycin administration significantly increased the percentage of neutrophils and basophils to the total WBCs as compared to the ketamine group, and the percentages of eosinophils and lymphocytes was decreased. On the contrary, the administration of wortmannin significantly increased the percentages of neutrophils, monocytes, and eosinophils to total WBCs as compared to the ketamine group, whereas the percentage of lymphocytes was decreased. Moreover, treatment with rapamycin alleviated the eosinophil-mediated inflammation to improve the pathogenesis of KIC.

Autophagy is a cellular protective process of damaged organelles, such as mitochondria and endoplasmic reticulum, which maintain cellular homeostasis. Autophagy plays a dual role in cancer: (A) suppressing tumor growth by decreasing damaged proteins and organelles accumulation, and (B) facilitates tumorigenesis by promoting cancer cell survival, proliferation, and tumor growth of established tumors [41]. Inhibition of the mTOR signaling pathway by rapamycin activates autophagy. Rapamycin can induce dysregulated mitochondrial biogenesis and apoptosis of tumor cells, and eventually reducing tumorigenesis [41,42,43]. In the present study, autophagy may be initiated via urinary metabolites (ketamine and norketamine). Rapamycin-induced autophagy might induce the therapeutic function of MSCs in wound healing via promoting suburothelial angiogenesis, and improve MSC-mediated vascularization in wound healing via regulation of VEGF secretion. Besides, treatment with wortmannin might promote angiogenesis, initiate wound healing, induce the releasing of VEGF, stimulate proliferation and differentiation of MSCs, reduce the level of oxidative stress, and result in the effect of tissue regeneration.

In this study, in the ketamine group, ketamine and its metabolites evoked m-TOR phosphorylation (8.0-fold) in the bladder compared to the control group. However, in the ketamine+rapamycin group, the levels of mTOR and p-mTOR protein were noticeably decreased compared to the ketamine group (Figure 4). These findings revealed that rapamycin inhibited ketamine-activated mTOR expression and elevated the level of autophagy-associated proteins. Liu et al. revealed that ketamine could dose-dependently increase the apoptosis of rat hippocampal neurons with p-mTOR upregulation and its downstream regulators (p-4E-BP-1 and p-p70S6K). Moreover, ketamine-induced apoptosis in hippocampal neurons was reversed meaningfully treated by rapamycin. Therefore, inhibition of the mTOR signaling pathway by rapamycin protected rat hippocampal neurons from ketamine-induced injuries via reducing apoptosis, oxidative stress, as well as Ca^2+^ concentration [44]. Moreover, some studies indicated that rapamycin significantly reduced neuronal death in the cerebral ischemia of rat mode [45]. Furthermore, inhibition of the mTOR signaling pathway by rapamycin prevented cytochrome *c* release to reduce brain damage in a cerebral ischemic rat model [26]. Rapamycin is an mTOR inhibitor to induce autophagy and increase the p38 phosphorylation, but inhibits the phosphorylation of Erk1/2 and Akt.

## 5. Conclusions

Treatment with ketamine significantly resulted in bladder overactivity, enhancement of interstitial fibrosis, impairment of endothelium, inhibition of angiogenesis, and elevation of the phosphorylation of Akt. While treatment with rapamycin alleviated the eosinophil-mediated inflammation to improve the pathogenesis of KIC, treatment with wortmannin reduced basophil-mediated inflammatory response. The application of rapamycin caused an inhibitory effect on vascular formation triggering by ketamine metabolite through PI3K/Akt/mTOR pathway, decreased the eosinophil-mediated inflammation, and ameliorated bladder hyperactivity, which improves bladder function in KIC. Treatment with wortmannin improved bladder angiogenesis and blood flow by increasing capillary density and VEGF expression to repair KIC.

## Figures and Tables

**Figure 1 biology-10-00488-f001:**
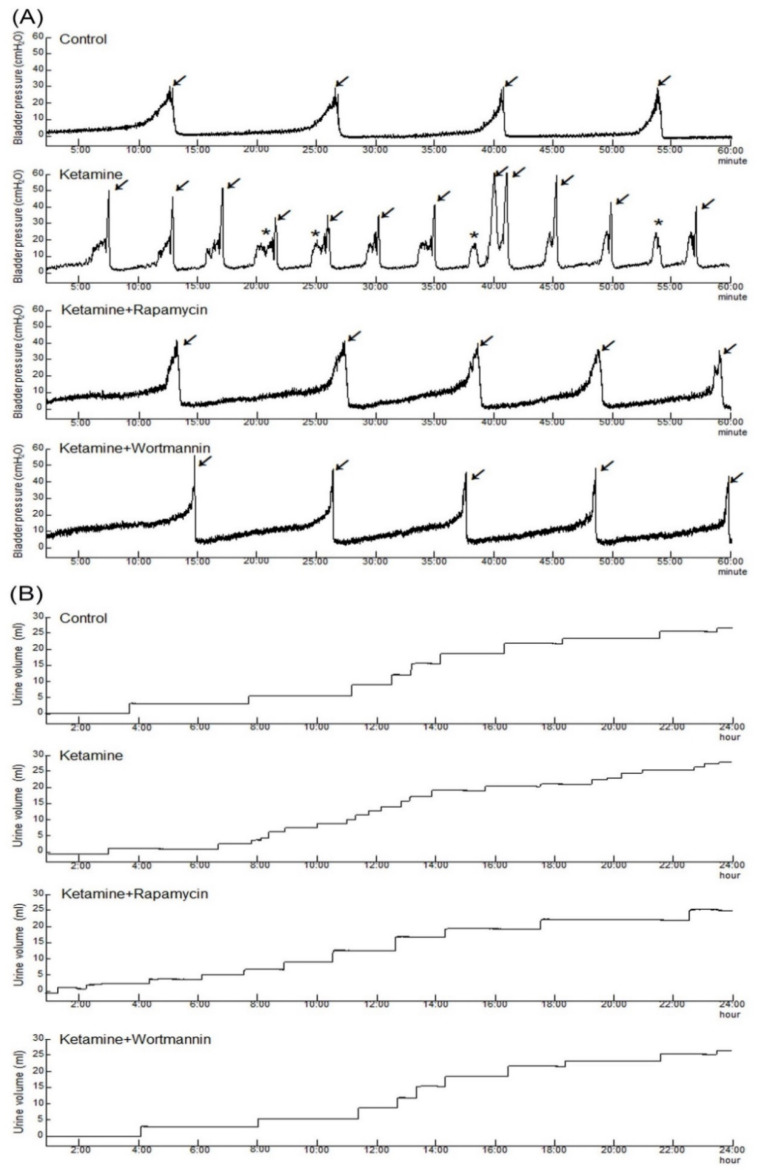
Bladder cystometric parameters and voiding behavior were shown in a KIC rat model. (**A**) Cystometry recordings of micturition pressure, voiding volume, and frequency, including voiding contraction (arrows) and non-voiding contraction (asterisks) in 60 min. (**B**) Tracing analysis of 24 h voiding behavior by metabolic cage, illustrating that the ketamine group significantly increased bladder maturation pressure, voiding contractions, non-voiding contractions, and micturition frequency. However, such increases were significantly reduced in the ketamine+rapamycin group and the ketamine+wortmannin group.

**Figure 2 biology-10-00488-f002:**
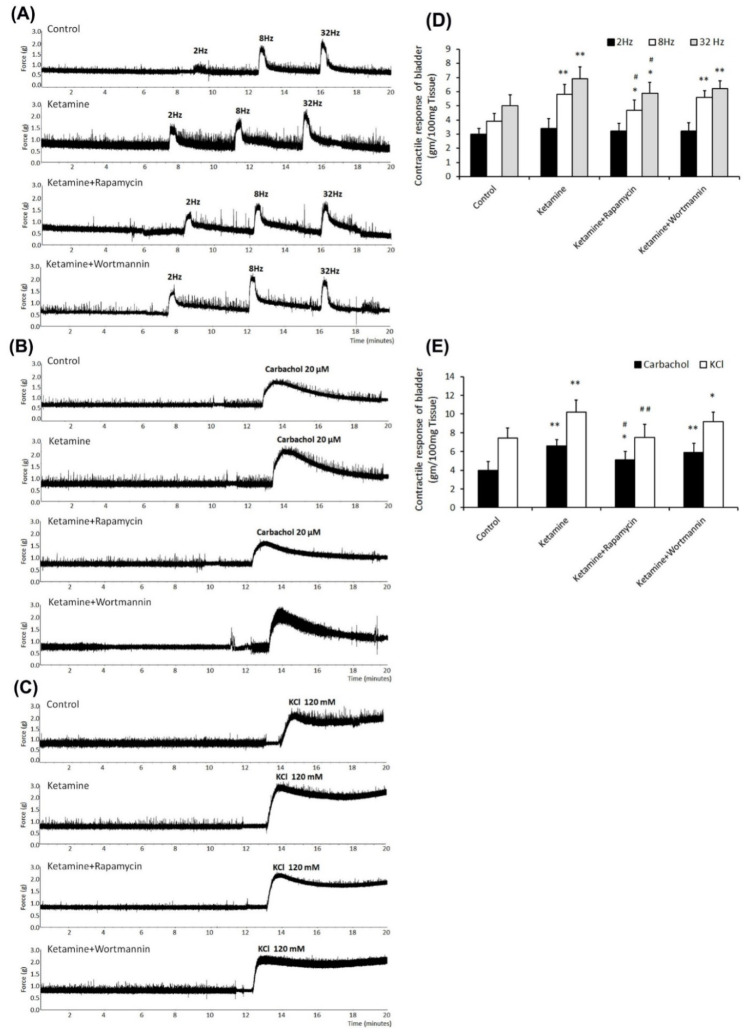
Contractile responses of bladder strips to electrical field, carbachol, and KCl stimulation. (**A**–**C**) Bladder contractile responses as examined by electrical field stimulation (2 Hz, 8 Hz, and 32 Hz) (**A**), carbachol (**B**), and KCl (**C**). (**D**,**E**) Statistics data of bladder contractile in responses to electrical field stimulation (**D**) or carbachol and KCl (**E**) for four different groups. The ketamine treatment caused significant contractile hyperactivity, whereas ketamine combined with rapamycin treatment improved these changes. Values are means ± SEM for n = 6. * *p* < 0.05 and ** *p* < 0.01 vs. the control group. ^#^
*p* < 0.05 and ^##^
*p* < 0.01 vs. the ketamine group.

**Figure 3 biology-10-00488-f003:**
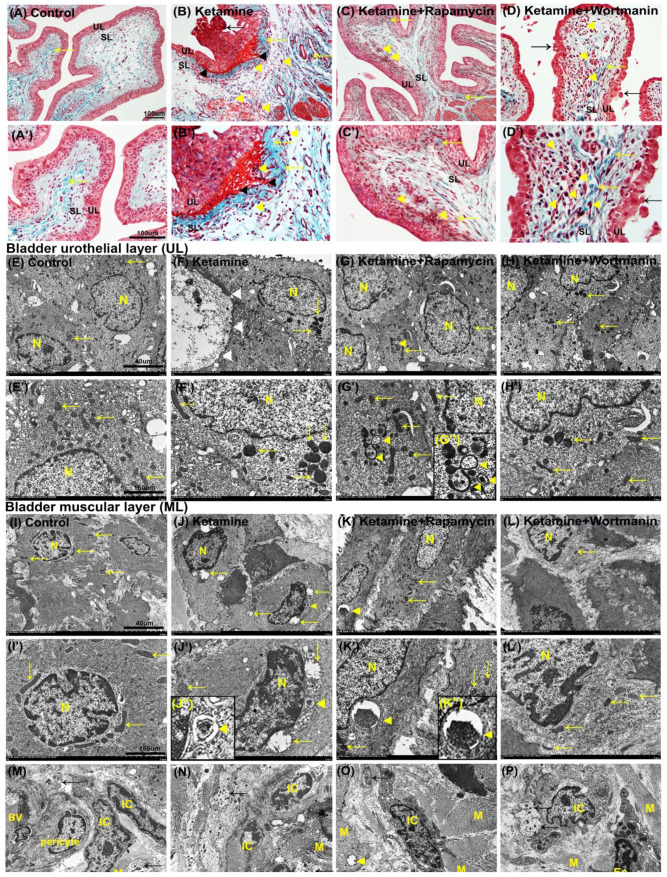
Histopathological examination for bladder damage as shown by Masson’s trichrome staining and ultrastructure observation detected by transmission electron microscope (TEM). (**A**–**D**,**A’**–**D’**) The bladder pathological features of the control group (**A**,**A’**), the ketamine group (**B**,**B’**), the ketamine+rapamycin group (**C**,**C’**), and the ketamine+wortmannin group (**D**,**D’**). Masson’s trichrome stain showed the blue-stained collagen, and the red-counterstained DSM highlighted for each image. In the control group (**A**,**A’**), there were three to five layers of the urothelial layer (UL), and only sparse collagen (yellow arrow) distributed in the suburothelial layer (SL). In the ketamine group (**B**,**B’**), the staining data showed denuded urothelial mucosa (black arrow), erythrocyte debris under the urothelium (black arrowheads), mononuclear cell infiltration (yellow arrowheads), and increased bladder fibrosis (yellow arrows). In contrast, morphological evaluation of the ketamine+rapamycin group improved ketamine-induced bladder damage by reducing ulcerated urothelium, mononuclear cell infiltration (yellow arrowheads), and interstitial fibrosis (yellow arrow) (**C**,**C’**). Moreover, the denuded urothelial mucosa (black arrow), mononuclear cell infiltration (yellow arrowheads), and increased interstitial fibrosis (yellow arrows) were shown in the ketamine+wortmannin (**D**,**D’**). (**E**–**L**,**E’**–**L’**) The ultrastructure of UL (**E**–**H**,**E’**–**H’**) or ML (**I**–**L**,**I’**–**L’**) was examined under TEM in the control group (**E**,**E’**,**I**,**I’**), the ketamine group (**F**,**F’**,**J**,**J’**), the ketamine+rapamycin group (**G**,**G’**,**K**,**K’**) and the ketamine+wortmannin group (**H**,**H’**,**L**,**L’**). The enlarged images for autophagosome and autolysosome in 3G’’, 3J’’ and 3K’’ were inserted in the bottom. (**M**–**P**) The ultrastructure of interstitial cells (ICs) located in close proximity to the muscle cells (M), nerve ending between smooth muscle bundles detected by TEM. Yellow arrows indicated the mitochondria. Black arrows denoted nerve bundle (nerve ending). Yellow arrowheads indicated organelle engulfed by autophagosome or autolysosome. Many damaged or swelling mitochondria and damaged organelle were engulfed by double-membrane autophagosomes in the ketamine+rapamycin group. Scale bar = 40 μm (Magnification ×4000) and 160 μm (Magnification ×100,000). N, nucleus; M, muscle; BV, blood vessel; Eo, eosinophil; IC, interstitial cell.

**Figure 4 biology-10-00488-f004:**
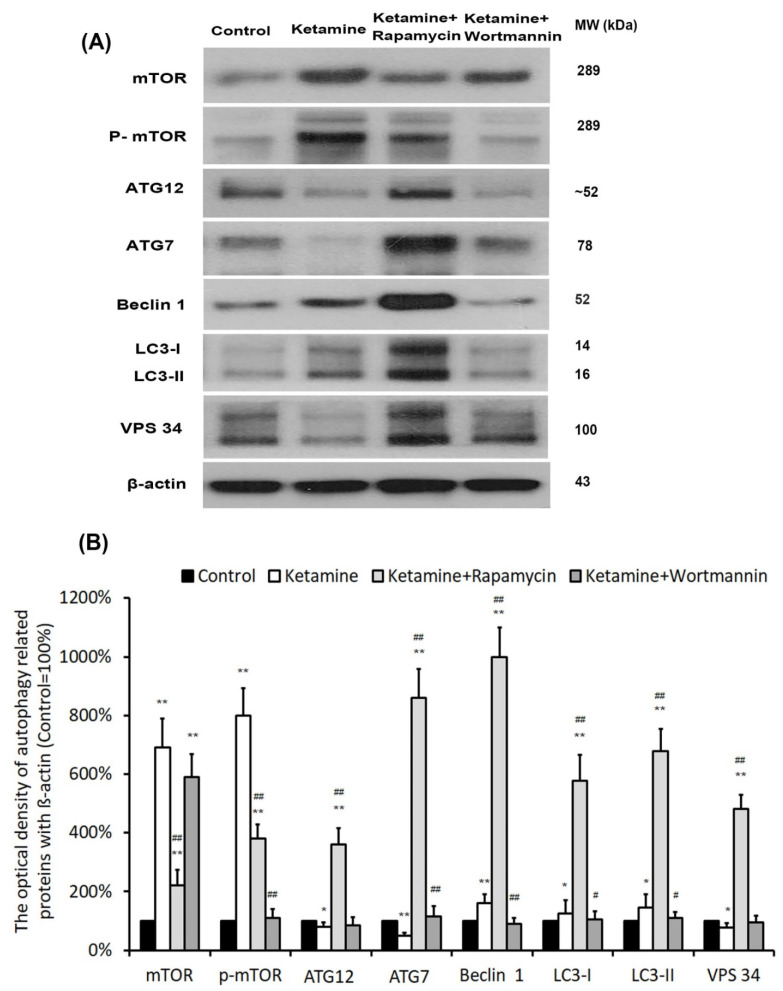
Treatment with rapamycin altered the expression of autophagy-associated proteins. (**A**) Representative western blots of autophagy-related genes are expressed: mTOR, mammalian target of rapamycin; ATG, autophagry-related protein; LC3, microtubule-associated protein 1A/1B-light chain 3; VPS 34, vacuolar protein sorting 34. (**B**) Protein expressions of autophagy-associated proteins were quantified against ß-actin. Results were normalized to the control group (100%). The protein level of autophagy-related genes was significantly increased in the ketamine+rapamycin group. However, this level was significantly decreased in the ketamine+wortmannin group. Values were presented as the mean ± SD, * *p* < 0.05, ** *p* < 0.01 versus control group; ^#^
*p* < 0.05, ^##^
*p* < 0.01 versus ketamine group.

**Figure 5 biology-10-00488-f005:**
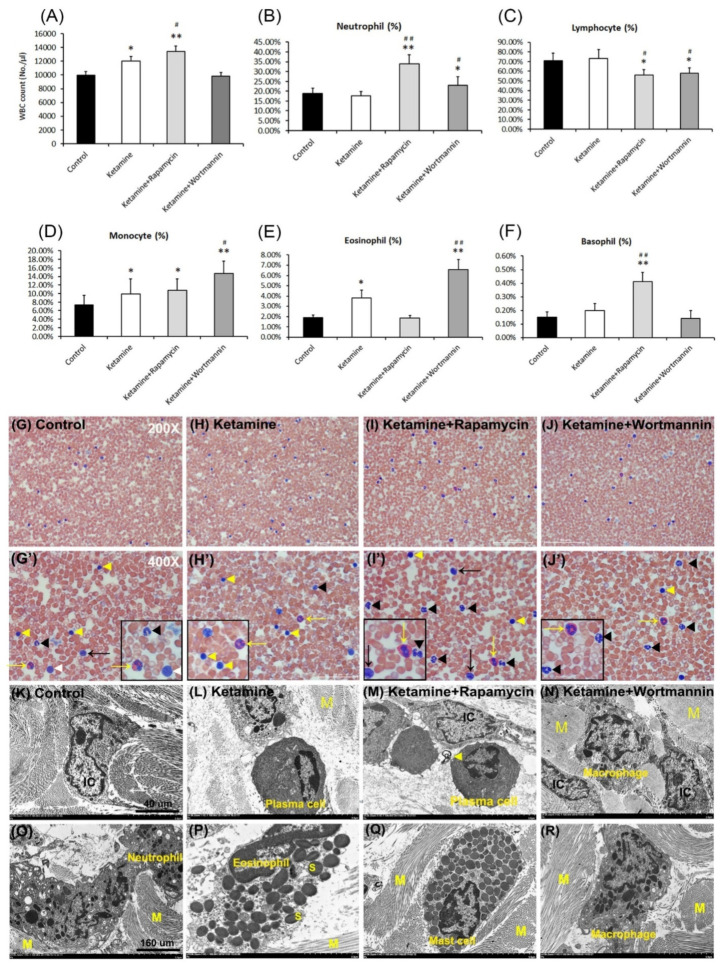
Significant changes in the numbers of total white blood cells (WBCs) and leukocyte differential counts. Quantification of the number of total WBCs was shown in (**A**). Quantifications for the percentage (%) of circulating neutrophils (**B**), lymphocytes (**C**), monocytes (**D**), eosinophils (**E**), and basophils (**F**) to total WBCs were present. Treatment with rapamycin alleviated eosinophil-mediated inflammation to improve the pathogenesis of KIC. Data are presented as means ± SD, * *p* < 0.05, ** *p* < 0.01 versus control group; ^#^
*p* < 0.05, ^##^
*p* < 0.01 versus ketamine group. (**G**–**J** and **G’**–**J’**) The prominent feature of circulating leukocytes stained with a blood smear. Enlarged images (**G’**–**J’**) for prominent feature of leuocytes from images **G**–**J** were shown. Yellow arrow and arrowhead indicate eosinophil and lymphocyte, respectively. The black arrow and arrowhead represent monocyte and neutrophil, respectively. The white arrowhead denotes basophil. The enlarged images were inserted in the bottom right. (**K**–**N**) The morphology of leukocytes detected by TEM in the control group (**K**), the ketamine group (**L**), the ketamine+rapamycin group (**M**), the ketamine+wortmannin group (**N**). (**O–R**) Infiltrated inflammatory cells, including lymphocytes, neutrophils (**O**), eosinophils (**P**), mast cells (**Q**), and macrophages (**R**), into lamina propia. Specific leucocytes in various sites of the bladder were reflected in increased numbers in the circulation. Scale bar = 40 μm (Magnification ×4000) and 160 μm (Magnification ×100,000). N, nucleus; M, muscle.

**Figure 6 biology-10-00488-f006:**
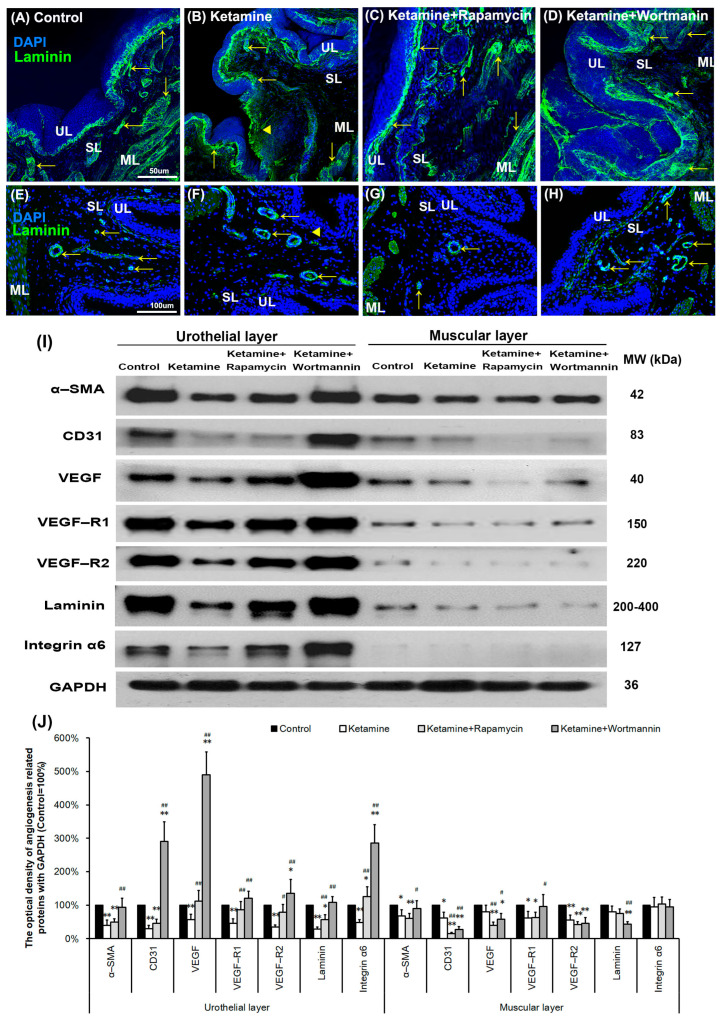
Autophagy altered bladder angiogenesis in the pathogenesis of KIC. (**A**–**H**) Immunofluorescence analysis of laminin, and α-SMA (**E**–**H**) (angiogenesis marker, arrows) expression after treatment with saline (**A**,**E**), ketamine (**B**,**F**), ketamine+rapamycin (**C**,**G**), and ketamine+wortmannin (**D**,**H**) in urothelial layer (UL), suburothelial layer (SL) and muscular layer (ML) of bladder tissue. In the ketamine+rapamycin group, bladder angiogenesis was mainly expressed in the marginal zone of lamina propria (suburothelial layer) near the urothelial basement membrane. The ketamine+wortmannin group significantly intensified the expression levels in the urothelial layer as compared to the other groups. The nuclei were counterstained by DAPI (blue). Yellow arrowheads indicate the denuded urothelial mucosa. Scale bar = 50 μm (Magnification ×200) and 100 μm (Magnification ×400). (**I**) Western blots of angiogenesis markers, including α-SMA, CD31 (endothelial marker), VEGF, VEGF-R1, VEGF-R2 (VEGF receptor), laminin, and integrin-α6 (laminin receptor), were quantified against glyceraldehyde-3-phosphate dehydrogenase (GAPDH). (**J**) Results were normalized to the control group (equal to 100%). α-SMA, alpha-smooth muscle actin; CD31, cluster of differentiation 31; VEGF, vascular endothelial growth factor. Data are expressed as means ± SD for n = 6, * *p* < 0.05, ** *p* < 0.01 versus the control group; ^#^
*p* < 0.05, ^##^
*p* < 0.01 versus the ketamine group.

**Figure 7 biology-10-00488-f007:**
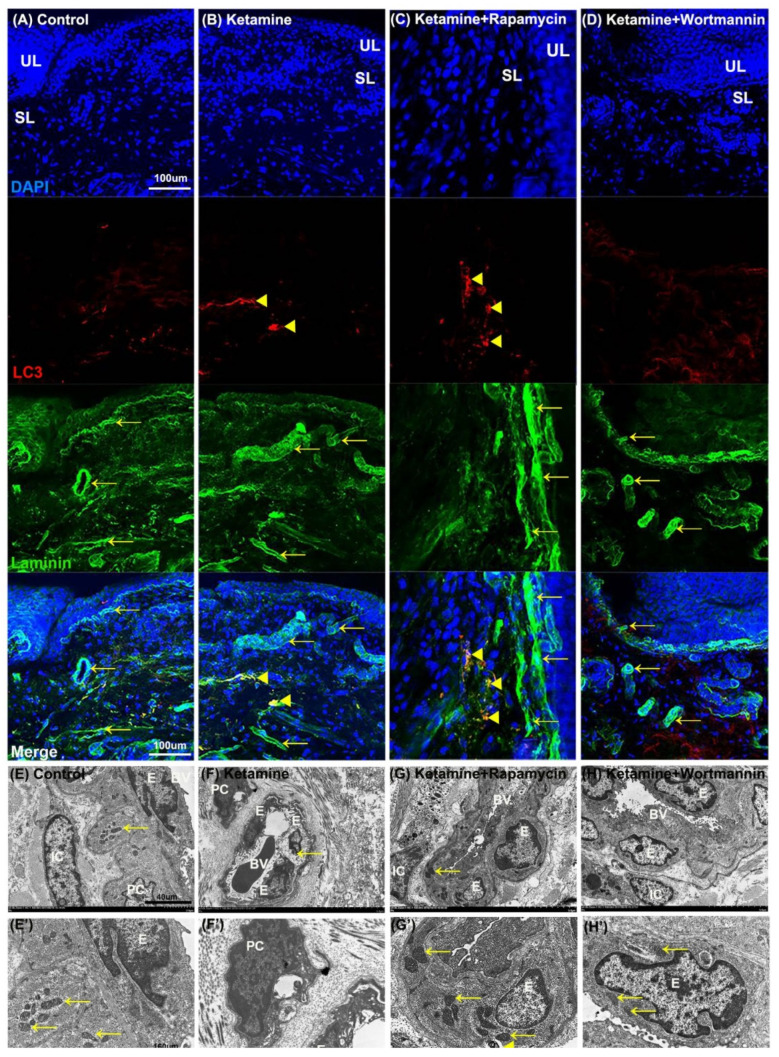
The relationship between autophagy and angiogenesis in the pathogenesis of KIC was evaluated. (**A**–**D**) Immunofluorescence analysis of Laminin (green) and LC3 (red) expression after treatment with saline (**A**), ketamine (**B**), ketamine+rapamycin (**C**), or ketamine+wortmannin (**D**) in urothelial layer (UL), and suburothelial layer (SL) of bladder tissue. Double immunostaining of laminin (green, yellow arrows, upper right panels) and LC3 (red, yellow arrowheads, bottom left panels) was shown in UL and SL. Nuclear DNA was labeled with DAPI (blue). The merged image from the bottom right panels (yellow) was shown. The LC3 (arrowheads) expression in the SL was identified in the ketamine+rapamycin group. Scale bar = 100 μm. (**E**–**H** and **E’**–**H’**) The ultrastructure of the bladder blood vessel was examined under TEM. Enlarged images (**E’**–**H’**) for prominent ultrastructure of blood vessel from images (**E’**–**H’**) were shown. There was autolysosome in the endothelium of the ketamine+rapamycin group. Scale bar = 40 μm (Magnification ×4000) and 160 μm (Magnification ×100,000). Yellow arrows indicated the mitochondria. Yellow arrowheads denoted autolysosome. E, endothelium; IC, interstitial cell; BV, blood vessel; PC, pericyte; LC3, microtubule-associated protein 1 light chain 3.

**Figure 8 biology-10-00488-f008:**
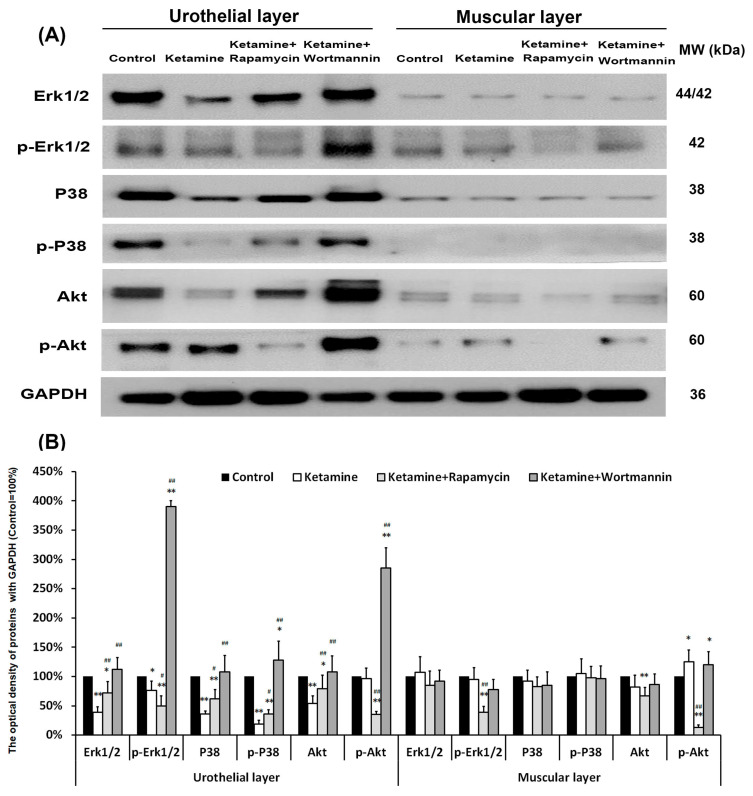
Western blots of signaling pathway kinase, including Erk1/2, p-Erk1/2, P38, p-P38, Akt, and p-Akt, were quantified against GAPDH (**A**). (**B**) Results were normalized to the control group (equal to 100%). The data revealed that treatment with ketamine significantly increased the Akt phosphorylation, but reduced the phosphorylation of Erk1/2 and p38 as compared to the control group. However, treatment with rapamycin increased the p38 phosphorylation, but suppressed the expression of p-Erk1/2 and p-Akt. Data were expressed as means ± SD for n = 6, * *p* < 0.05, ** *p* < 0.01 versus the control group; ^#^
*p* < 0.05, ^##^
*p* < 0.01 versus the ketamine group.

**Figure 9 biology-10-00488-f009:**
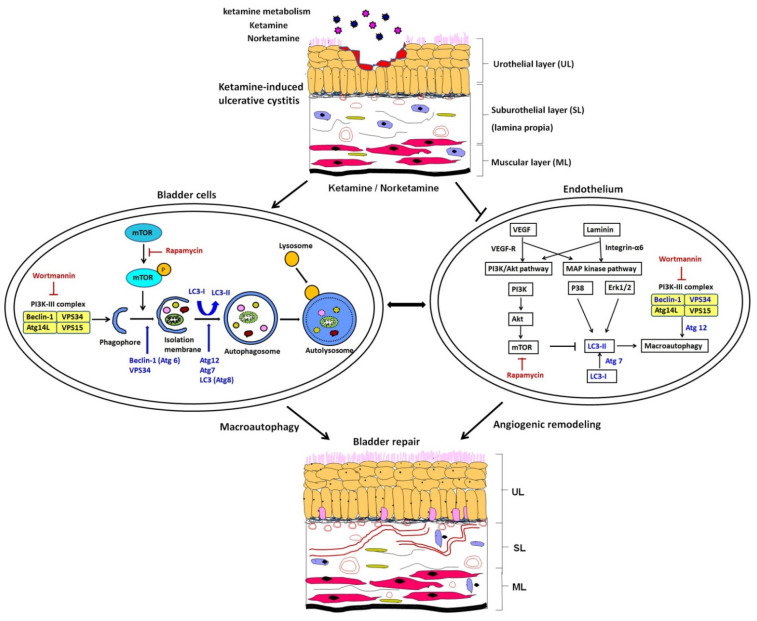
Representative image for the anti-angiogenesis potential of autophagy. The proposed potential mechanism of anti-angiogenesis was induced by ketamine and its metabolite through PI3K/Akt/mTOR pathway**,** leading to contribution to the pathogenesis of KIC. Our observations implied that treatment with ketamine significantly resulted in bladder overactivity, enhanced interstitial fibrosis, impaired endothelium, inhibited angiogenesis, and elevated the phosphorylation of Akt. However, these changes were ameliorated in the ketamine+rapamycin group. The application of rapamycin caused an inhibitory effect on vascular formation, removed ketamine metabolites, decreased the inflammation, and ameliorated bladder hyperactivity, leading to improving bladder function in KIC, while wortmannin improved bladder angiogenesis by increasing capillary density and VEGF expression, to reverse anti-angiogenic effect to repair KIC.

**Table 1 biology-10-00488-t001:** General characteristics, urodynamic parameters, contractile responses, and count analysis of leukocytes for different experimental groups.

	Control	Ketamine	Ketamine+ Rapamycin	Ketamine+ Wortmannin
No. rats	10	10	10	10
General characteristics				
Body weight (g)	248.4 ± 38.2	231.7 ± 46.8	240.5 ± 23.8	237.5 ± 26.4
Bladder weight (mg)	141.3 ± 16.8	173.5 ± 18.7 *	155.0 ± 10.3 ^#^	167.0 ± 14.9 *
Water intake (cc/24 h)	30.3 ± 5.9	29.9 ± 6.4	35.8 ± 7.3	37.4 ± 7.1
Urine output (cc/24 h)	19.7 ± 5.6	20.7 ± 5.3	24.3 ± 5.8	22.9 ± 4.7
Serum parameters				
Ketamine (ng/mL)	ND	ND	ND	ND
Norketamine (ng/mL)	ND	ND	ND	ND
Urine parameters				
Ketamine conc. (ng/mL)	ND	1268.3 ± 231.5 **	1096.7 ± 175.8 **	1108.7 ± 195.4 **
Norketamine conc. (ng/mL)	ND	10150.2 ± 1146.8 **	8790.0 ± 1235.4 **	9690.0 ± 1033.7 **

Note: No., number; ND, not detected; Values are mean ± SD for n = 10. * *p* < 0.05; ** *p* < 0.01 vs. control (saline) group; ^#^
*p* < 0.05 vs. ketamine group.

**Table 2 biology-10-00488-t002:** Urodynamic parameters, contractile responses, and count analysis of leukocytes for different experimental groups.

	Control	Ketamine	Ketamine+ Rapamycin	Ketamine+ Wortmannin
No. rats	10	10	10	10
Urodynamic parameters				
Frequency (No. voids/24 h)	12.2 ± 2.7	30.8 ± 7.6 **	13.8 ± 3.5 ^##^	14.8± 3.3 ^##^
Peak micturition pressure (cmH_2_O)	32.6 ± 3.2	48.6 ± 5.5 **	39.7 ± 3.8 * ^#^	40.7 ± 5.8 * ^#^
Voided volume (mL)	1.79 ± 0.29	1.06 ± 0.27 *	1.59 ± 0.40 ^#^	1.48 ± 0.32 ^#^
No. non-voiding contractions between micturition (No./60 min)	0	3.7 ± 0.6 **	0 ^#^	0 ^#^
Contractile responses				
Electrical-field stimulation (2 Hz)	3.0 ± 0.42	3.4 ± 0.69	3.2 ± 0.58	3.2 ± 0.60
Electrical-field stimulation (8 Hz)	3.9 ± 0.58	5.8 ± 0.70 **	4.7 ± 0.71 * ^#^	5.6 ± 0.46 **
Electrical-field stimulation (32 Hz)	5.0 ± 0.79	6.9 ± 0.86 **	5.9 ± 0.74 * ^#^	6.2 ± 0.59 **
Carbachol (20 μM)	4.0 ± 0.96	6.6 ± 0.64 **	5.1 ± 0.90 * ^#^	5.9 ± 0.94 **
KCl (120 mM)	7.4 ± 1.10	10.2 ± 1.30 **	7.5 ± 1.40 ^##^	9.2 ± 1.00 *
Count analysis of leukocytes				
Total white blood cells (No./μL)	9936.5 ± 542	12014 ± 638 *	13419 ± 782 ** ^#^	9820 ± 562
Neutrophils (%)	18.77 ± 2.78%	17.70 ± 2.24%	33.86 ± 4.78% ** ^##^	22.97 ± 4.28% * ^#^
Lymphocytes (%)	71.00 ± 7.66%	72.96 ± 9.36%	56.10 ± 5.37% * ^#^	57.72 ± 5.61% * ^#^
Monocytes (%)	7.33 ± 2.20%	9.93 ± 3.52% *	10.71 ± 2.69% *	14.74 ± 2.82% ** ^#^
Eosinophils (%)	1.90 ± 0.27%	3.82 ± 0.74% *	1.85 ± 0.26% ^#^	6.58 ± 0.99% ** ^##^
Basophils (%)	0.15 ± 0.04%	0.20 ± 0.05%	0.41 ± 0.07% * ^##^	14 ± 0.06%

Note: No., number; ND, not detected; Values are mean ± SD for n = 10. * *p* < 0.05; ** *p* < 0.01 vs. control (saline) group; ^#^
*p* < 0.05; ^##^
*p* < 0.01 vs. ketamine group.

## Data Availability

Not applicable.

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
