# Peer review of "Autophagy Alters Bladder Angiogenesis and Improves Bladder Hyperactivity in the Pathogenesis of Ketamine-Induced Cystitis in a Rat Model"

_biology, 2021, doi:10.3390/biology10060488_

Round 1
Reviewer 1 Report
Lu et al., have studied the “Autophagy alters bladder angiogenesis and improves bladder hyperactivity in the pathogenesis of ketamine-induced cystitis in a rat model”. This an excellent study. This is a massive work that the authors have conducted to determine autophagy's role in the ketamine-induced cystitis model. The use of both autophagy inducer and inhibitor are very reasonable. Material and methods are well described, and an illustration representing the global view of the proposed mechanism is impeccable.
I have some suggestions and queries.
Line 82: autophagy-related genes (Atg) ----- autophagy-related (ATG) genes
Line 104: Check the sentence, inhibitor of mTOR should activate autophagy.
Line 147: 09% ----- 0.9%
Many aspects of manuscript is not uniform, e.g., 2h --- 2 h (line 149); blood --- blood (line 211); 24hrs --- 24 h (table 1); 60mins ---- 60 min (table 1); P < 00.5 or P<0.05, and many other places. Please correct and proof-read one more time.
Figure 3: Autophagosomes and autolysosomes are not visible; it would be great if authors enlarge each organelle.
Figure 5: I am wondering how authors have differentiated the plasma cell or macrophages.
Figure 7 (A-D): It would be nice if the authors represent the panels in the parallel mode. LC3 should appear like a punctae formation, whereas punctae is not legible in the images. https://doi.org/10.3390/cells9051321
Figure 9: The step “Conversion of LC3-I to LC-3-II” should be moved to one strep ahead.
Lysosome and autolysosome should be separate entities. Rapamycin inhibits the mTOR; in urothelium this representation is lacking. Please see this reference for global view https://doi.org/10.1038/s41580-018-0033-y.
Line 80-84: This citation would fit here. https://doi.org/10.3390/cells9051321
Author Response
Dear Dr. Diao and Dr.Review1,
We wish to thank the Editorial Board for the review of our manuscript entitled " Autophagy alters bladder angiogenesis and improves bladder hyperactivity in the pathogenesis of ketamine-induced cystitis in a rat model", which is being considered by the Biology for publication.
By addressing every comment made by the reviewers, we have revised our manuscript (Please see the attachment). All the changes made in the manuscript are marked in red font. We would like to thank you and the Editorial Board for the consideration and the intelligent review of our manuscript, which results in the revised manuscript of better quality.
Sincerely yours,
Yung-Shun Juan MD, PhD.
May 22nd, 2021
Reviewer comments:
Review 1:
Comments and Suggestions for Authors
Lu et al., have studied the “Autophagy alters bladder angiogenesis and improves bladder hyperactivity in the pathogenesis of ketamine-induced cystitis in a rat model”. This an excellent study. This is a massive work that the authors have conducted to determine autophagy's role in the ketamine-induced cystitis model. The use of both autophagy inducer and inhibitor are very reasonable. Material and methods are well described, and an illustration representing the global view of the proposed mechanism is impeccable.
1) Line 82: autophagy-related genes (Atg) ----- autophagy-related (ATG) genes
Response: As suggested by the reviewer, we have corrected the “autophagy-related genes (Atg)” to “autophagy-related (ATG) genes” in manuscript (please refer to page 3, line 84; page 6, lines 234-237 and 257-258; page 16, lines 445-446 and 450; page 17, line 457; page 28, line 667; page 17, Figure 4).
(Figure 4)
2) Line 104: Check the sentence, inhibitor of mTOR should activate autophagy.
Response: Yang et al. showed that rapamycin (an inhibitor of mTOR) treatment reduced the autophagy markers (LC3-II and Beclin-1) levels to equal or below the baseline levels in both the cortex and hippocampus after ischemia and reperfusion for 7 days. Therefore, we rewrite the sentence “Inhibition of mTOR pathway by rapamycin prevents cytochrome c release and reduces ischemic brain damage in rats subjected to transient forebrain ischemia” in the Introduction section, lines 106-108.
3) Line 147: 09% ----- 0.9%
Response: Thank you for your suggestion. We have corrected the “09%” to “0.9%” (please refer to page 4, line 150, Materials and methods section 2.1).
4) Many aspects of manuscript is not uniform, e.g., 2h --- 2 h (line 149); blood --- blood (line 211); 24hrs --- 24 h (table 1); 60mins ---- 60 min (table 1); P < 00.5 or P<0.05, and many other places. Please correct and proof-read one more time.
Response: As suggested, we have corrected and proof-read the whole manuscript carefully. For example, Materials and methods section (please refer to page 4, lines 138, 152 and 170; page 5, line215; page 6, line271), Figure legend section (please refer to page 8, line 306, in Figure 1; page 13, line 345 in Figure 2; page 18, line 461-462 in Figure 4; page 20, line 515-516 in Figure 5; page 22, lines 549-550 in Figure 6; page 26, line 623 in Figure 8), Table 1 (page 9, lines 311-312) and Table 2 (page 10, lines 315-316).
5) Figure 3: Autophagosomes and autolysosomes are not visible; it would be great if authors enlarge each organelle.
Response: As suggested, we have enlarged the cellular autophagosomes and autolysosomes in Figures 3G’, 3J’ and 3K’. The enlarged images, including 3G’’, 3J’’ and 3K’’ were inserted in the bottom right and supplementary note added in Figure legend section (please refer to page 15, line 392).
6) Figure 5: I am wondering how authors have differentiated the plasma cell or macrophages.
Response: According to the ultrastructure of WBCs by blood smear and TEM, plasma cells are characterized by an eccentric round nucleus with condensed chromatin and a well-developed nucleolus. However, macrophages are large, round cell with a central round nucleus and abundant clear, often vacuolated cytoplasm. Numerous small pseudopodia extend from the cell, reflecting phagocytic ability and amoeboid movement by TEM. We also added the descriptions in the Result section (please refer to page 18, line 503-506), as” plasma cell (an eccentric round nucleus with condensed chromatin and a well-developed nucleolus) (Figure 5M), macrophage (a large, round cell with a central round nucleus, numerous small pseudopodia extend from the cell and vacuolated cytoplasm) (Figures 5N and 5R).”
7) Figure 7 (A-D): It would be nice if the authors represent the panels in the parallel mode. LC3 should appear like a punctae formation, whereas punctae is not legible in the images. https://doi.org/10.3390/cells9051321
Response: As suggested by the reviewer, we have represented the Figure 7 panels in the parallel mode. Generally, LC3 appeared like a punctae formation. However, the present study using whole mount bladder for confocal laser scanning microscopy. The reason why the punctae formation in Figure 7 was not clear may be due to the whole mount is too thick (25 μm), which makes the LC3 visual image more vague.
8) Figure 9: The step “Conversion of LC3-I to LC-3-II” should be moved to one step ahead.
Response: We have moved the step “Conversion of LC3-I to LC-3-II” one step ahead in Figure 9 as suggested. (please refer to page 27)
9) Lysosome and autolysosome should be separate entities. Rapamycin inhibits the mTOR; in urothelium this representation is lacking. Please see this reference for global view https://doi.org/10.1038/s41580-018-0033-y.
Response: As suggested by the reviewer, lysosome and autolysosome are separate entities. According to several studies, rapamycin inhibits the mTOR expression. However, in our study, ketamine and its metabolites evoked m-TOR phosphorylation in bladder of the ketamine group as comparison with the control group by western blot analysis. Besides, the levels of mTOR and p-mTOR protein in the ketamine + rapamycin group were noticeably declined as comparison with the ketamine group (Figure 4).
Thank you for your suggestion. We have corrected “urothelium” to “bladder cells” in Figure 9 and modified our result in Figure 9 shown below. According to the data of immunostaining and TEM in this study, autophagosomes were shown in urothelium, smooth muscle cell and endothelium in the bladder tissue.
Liu et al. revealed that ketamine could dose-dependently promote the apoptosis of rat hippocampal neurons with upregulation of p-mTOR and its downstream regulators (p-4E-BP-1 and p-p70S6K). Moreover, ketamine-induced apoptosis in hippocampal neurons was reversed significantly by the administration of rapamycin. Therefore, inhibition of mTOR signaling pathway by rapamycin treatment protected rat hippocampal neurons from ketamine-induced injuries via reducing apoptosis, oxidative stress, as well as Ca2+ concentration.( Neurol Res. 2019 Jan;41(1):77-86. ) . We have added the reference in the Discussion section (please refer to page 29, lines 770-772; page 30, lines 773-781).
10) Line 80-84: This citation would fit here. https://doi.org/10.3390/cells9051321
Response: As suggested by the reviewer, we have added the reference (https://doi.org/10.3390/cells9051321) in the Introduction section (please refer to page 3, line 86).

Reviewer 2 Report
Long term effect of ketamine on bladder and urinary system is known but the mechanism of this effect remains clear. I think this paper tried to explain some missing link. Lot of data in this article and that could be organized better for readability.
Minor:
1) Check line concentration in line 147.
2) Table could be divided in two.
3) Histological data could be presented with better clarity.
4) Figures could be separated too.
Major:
1) Result should be described in detail using quantitative data.
2) I think authors should mention and discuss bladder pressure and compare treatments with control.
3) Data which suggests little effect for ketamine + rapamycin could be submitted as a supplementary figure.
4) Author should discuss rapamycin role in cancer. Blocking mTOR by rapamycin attenuates bladder hyperactivity and pain.
Author Response
Dear Dr. Diao and Dr. Review2,
We wish to thank the Editorial Board for the review of our manuscript entitled " Autophagy alters bladder angiogenesis and improves bladder hyperactivity in the pathogenesis of ketamine-induced cystitis in a rat model", which is being considered by the Biology for publication.
By addressing every comment made by the reviewers, we have revised our manuscript (Please see the attachment). All the changes made in the manuscript are marked in red font. We would like to thank you and the Editorial Board for the consideration and the intelligent review of our manuscript, which results in the revised manuscript of better quality.
Sincerely yours,
Yung-Shun Juan MD, PhD.
May 22nd, 2021
Review 2:
Comments and Suggestions for Authors
Long term effect of ketamine on bladder and urinary system is known but the mechanism of this effect remains clear. I think this paper tried to explain some missing link. Lot of data in this article and that could be organized better for readability.
Minor:
1) Check line concentration in line 147.
Response: Thank you for your suggestion. We have corrected the “09%” to “0.9%” (please refer to page 4, line 150, Materials and methods section 2.1).
2) Table could be divided in two.
Response: We have divided the Table 1 in two as suggested. We also added the descriptions in the Result section (please refer to page 9 and 10; page 12, result section 3.3, line 350-356).
3) Histological data could be presented with better clarity.
Response: As suggested by the reviewer, we have presented the histological data with better clarity in 1200dpi.
4) Figures could be separated too.
Response: Thank you for your suggestion. We have separated and presented the figure 7 in parallel models (please refer to page 23). However, in order to simply the figures, it is hard to separate the figures.
Major:
1) Result should be described in detail using quantitative data.
Response: Response: As suggested by the reviewer, quantification data have been added in Results section 3.6 (page 16, lines 448-451), 3.8 (page 22, lines 555-561 and 564-567), and 3.10 (page 24, line 614; page 26, line 627 and 632-633).
2) I think authors should mention and discuss bladder pressure and compare treatments with control.
Response: As suggested by the reviewer, we have added more information on bladder pressure and compare treatments with control in the Discussion section (please refer to page 28, lines 690-694) as “However, the peak micturition pressures in the ketamine treatment groups are still higher than the control. This might imply in addition to bladder cell autophage changes, there were some other factors affecting bladder micturition pressure, such as pelvic floor hyertonicity induced bladder outlet obstruction or neuroinflammation. ”
3) Data which suggests little effect for ketamine + rapamycin could be submitted as a supplementary figure.
Response: Thank you for your suggestion. However, we believed it is important to present the treatment effect of ketamine + rapamycin group for the readers a global view of the autophage alternation on bladder hyperactivity.
4) Author should discuss rapamycin role in cancer. Blocking mTOR by rapamycin attenuates bladder hyperactivity and pain.
Response: As suggested by the reviewer, we have added more information on rapamycin role in cancer, and attenuates bladder hyperactivity and pain, in the Discussion section (please refer to page 29, lines 756-763), as ” Autophagy plays a dual role in cancer: (A) suppressing tumor growth by decreasing damaged proteins and organelles accumulation and (B) facilitates tumorigenesis by pro-moting cancer cell survival, proliferation and tumor growth of established tumors [40]. Inhibition of the mTOR signaling pathway by rapamycin activates autophagy. Rapamy-cin can induce dysregulated mitochondrial biogenesis and apoptosis of tumor cells, and eventually reducing tumorigenesis [40-42]. In the present study, autophagy may be initiated via urinary metabolite (ketamine and norketamine). Rapamycin-induced autophagy might ….”.

Round 2
Reviewer 1 Report
The authors have extensively revised the manuscript by considering the reviewer's comments.
The present manuscript is acceptable for publication. Herewith I am endorsing the manuscript for publication.
Thank you very much